# Moso Bamboo Invasion Reshapes Community Structure of Denitrifying Bacteria in Rhizosphere of *Alsophila spinulosa*

**DOI:** 10.3390/microorganisms10010180

**Published:** 2022-01-14

**Authors:** Youwei Zuo, Huanhuan Qu, Changying Xia, Huan Zhang, Jiahui Zhang, Hongping Deng

**Affiliations:** 1Center for Biodiversity Conservation and Utilization, School of Life Sciences, Southwest University, Beibei, Chongqing 400715, China; zuoyw1995@email.swu.edu.cn (Y.Z.); qhh79196625@163.com (H.Q.); xiachangying1995@163.com (C.X.); zh4433@email.swu.edu.cn (H.Z.); zjhscxc@swu.edu.cn (J.Z.); 2Chongqing Key Laboratory of Plant Resource Conservation and Germplasm Innovation, Institute of Resources Botany, School of Life Sciences, Southwest University, Beibei, Chongqing 400715, China; 3Chongqing Academy of Science and Technology, Low Carbon and Ecological Environment Protection Research Center, Liangjiang New Area, Chongqing 401123, China

**Keywords:** *Phyllostachys pubescens*, *Alsophila spinulosa*, invasion, denitrifying bacteria, nitrogen cycle

## Abstract

The uncontrolled invasion of moso bamboo (*Phyllostachys pubescens*) dramatically alters soil nitrogen cycling and destroys the natural habitat of *Alsophila spinulosa*. Nevertheless, no clear evidence points out the role of denitrifying bacteria in the invasion of bamboo into the habitat of *A. spinulosa*. In the present study, we found that low (importance value 0.0008), moderate (0.6551), and high (0.9326) bamboo invasions dramatically altered the underground root biomass of both *P. pubescens* and *A. spinulosa*. The root biomass of *A. spinulosa* was maximal at moderate invasion, indicating that intermediate disturbance might contribute to the growth and survival of the colonized plant. Successful bamboo invasion significantly increased rhizospheric soil available nitrogen content of *A. spinulosa*, coupled with elevated denitrifying bacterial abundance and diversity. *Shewanella*, *Chitinophaga*, and *Achromobacter* were the primary genera in the three invasions, whereas high bamboo invasion harbored more denitrifying bacteria and higher abundance than moderate and low invasions. Further correlation analysis found that most soil denitrifying bacteria were positively correlated with soil organic matter and available nitrogen but negatively correlated with pH and water content. In addition, our findings illustrated that two denitrifying bacteria, *Chitinophaga* and *Sorangium,* might be essential indicators for evaluating the effects of bamboo invasion on the growth of *A. spinulosa*. Collectively, this study found that moso bamboo invasion could change the nitrogen cycling of colonized habitats through alterations of denitrifying bacteria and provided valuable perspectives for profound recognizing the invasive impacts and mechanisms of bamboo expansion.

## 1. Introduction

Moso bamboo (*Phyllostachys pubescens* Mazel ex J. Houz.; synonym *P. edulis* (Carrière) J. Houz) is a member of the Poaceae and tree-like woody bamboo, harboring crucial ecological, practical, and cultural values [1,2]. *P. pubescens* is able to grow in mountainous areas, possesses a rapid regrowth rate after harvesting, and its culms can reach over 25 m in height with a 20-cm basal diameter [3]. Nevertheless, *P. pubescens* is also notorious for its invasive characteristics that can, unbridled, invade natural forests throughout subtropical China [2]. The uncontrolled invasion of bamboo has exerted tremendous influences on colonized forests, such as changed forest structure and plant composition [4], decreased species biodiversity and ecological stability [5], altered soil nitrogen (N) pools and cycling rates [6,7], and reshaped microbial diversity and composition [8]. For instance, the bamboo invasion has been documented to promote the input of labile soil organic matter fraction and increase soil bacterial diversity after invading a Japanese cedar (*Cryptomeria japonica*) plantation [9,10].

Tree fern *Alsophila spinulosa* (Hook) Tryon, belonging to the family Cyatheaceae, is an endangered relic plant with a tree-like trunk [11,12]. *A. spinulosa* had been flourishing during the Mesozoic era; nonetheless, after being affected by glaciation, its distribution area declined dramatically. The species mainly grows in mountainous areas, occupying shady niches with acidic soil (pH 4.5~5.5) at 400~900 m [13]. Currently, due to some deficiencies (e.g., short spore life-span, relatively high temperature, humidity and illumination requirements, and primitive root system and transfusion tissue), its modern distribution is restricted to southern China (18.5° N~30.5° N), Japan, and southeastern Asia [14]. In addition, *A. spinulosa* has been growingly threatened by exotic plant invasion, especially in nature reserves with minor artificial disturbance. For example, the severe invasion of moso bamboo significantly decreased root biomass and altered the root morphological plasticity of *A. spinulosa* in the Chishui Alsophila National Nature Reserve [15]. Hence, evaluating the impact of invasion on local plants and investigating the invasive mechanisms are essential to protect the endangered plant.

N is one of the most critical components in the ecosystem and an essential nutrient for plant growth and survival. Several processes regulate the N cycle, one of which is denitrification, with the help of denitrifying bacteria (e.g., *Shewanella*, *Achromobacter*, and *Pseudoxanthomonas*) [16,17,18,19]. Unconstrained bamboo expansion is well-reported to alter soil N cycle in the colonized habitat. A previous study found that the invasion of *P. pubescens* elevated the soil labile N pool, as well as microbial biomass carbon [10]. Similarly, Liu et al. [20] and Fukushima et al. [21] observed that the content of soil total N content in the adjacent evergreen broadleaved forest was lower than that of the bamboo forest, and the invasion of *P. pubescens* altered the distribution pattern of N stored in plants and soil. Yet, little is known about the impacts of *P. pubescens* invasion on *A. spinulosa*, especially from the viewpoint of denitrification, which would be critical for future understanding of the invasive mechanisms of bamboo expansion.

Thus, our objective in this study was to explore the influences and mechanisms of *P. pubescens* invasion on the highly endangered plant *A. spinulosa*. Specifically, the primary focuses of this study aim to: (1) reveal how properties of soil colonized with *A. spinulosa* respond to bamboo expansion; (2) identify the dominant rhizosphere denitrifying bacteria community and explore how invasion reshapes the denitrifying bacterial community; (3) evaluate the relationships between the denitrifying bacterial community and soil properties and growth of *A. spinulosa*. To address these questions, forest composition, root biomass, and denitrifying bacterial composition were examined meticulously in low, moderate, and high *P. pubescens-*invasive forests in a natural ecosystem located in Chishui Alsophila National Nature Reserve, Guizhou Province, China.

## 2. Materials and Methods

### 2.1. Community Survey and Soil Samples Collection

Chishui Alsophila National Nature Reserve is located in Guizhou Province, China (105°57′54″~106°7′7″ E 28°20′19″~28°28′48″ N), characterized by steep slope, deep valley, and closed terrain. The protected area belongs to a mid-subtropical humid monsoon climate, with 17.7 °C annual average temperature, 1200~1300 mm annual average precipitation, and 90% relative humidity. The soil type where *A. spinulosa* is grown is purple soil with pH 4.5~5.5.

The importance value (IMP) was calculated by dividing the frequency of *P. pubescens* by the frequency of all species. In this study, three sampling groups were chosen based on the IMP (range from 0~1) of *P. pubescens*, and three replicates for each group were collected at low (IMP < 0.3), moderate (0.3 < IMP < 0.7), and high (IMP > 0.7) *P. pubescens-*invasive sites (Figure 1A–C). The detailed information related to sampling site and forest structure are listed in Table 1. The characteristics (e.g., height and diameter at breast height) of *A. spinulosa* were recorded as well. In addition, to evaluate the effects of invasion on the root system, root samples at depths of 10 cm, 20 cm, and 30 cm (three replicates) were collected from both *P. pubescens* and *A. spinulosa*, respectively. The biomass value of root samples was calculated based on the equation *B* = *W*/[π × (0.1/2)^2^]. B~biomass (g/m^2^), W~root dry weight (g), and 0.1~diameter of the drill (m).

In order to investigate the role of denitrifying bacteria on the invasion of *P. pubescens* into *A. spinulosa*, rhizospheric soil samples (three replicates per group, with depth of 0~20 cm) of *A. spinulosa* from three invasive sites were excavated and shaken carefully to separate the soil from the roots, as described previously [22]. Finally, a total of nine soil samples were collected from the three invasive groups. A portion of the soil samples was immediately stored at −80 °C for subsequent molecular analysis, whereas the remaining soil was used for soil physical and chemical characterization.

### 2.2. Soil Chemical and Physical Properties

Soil physical and chemical analyses for pH, OM, WC, and available nitrogen (AN) were performed as previously reported [23]. Specifically, after conventional treatment, soil WC was measured by drying method; pH was measured by potentiometric method; OM was determined by potassium dichromate and sulfuric acid digestion method; and AN was measured by the alkali hydrolysis diffusion method.

### 2.3. DNA Extraction, QnorB Gene Amplification, and MiSeq Sequencing

Soil samples (0.5 g) were used to extract microbial DNA according to the QIAmp DNA Stool Mini Kit. To explore the soil denitrifying bacteria communities at three invasive sites, the target fragment qnorB was amplified with barcoded 2F 5′-GGNCAYCARGGNTAYGA-3′ and 5R 5′-ACCCANAGRTGNACNACCCACCA-3′. PCR amplification was conducted in an eight-cycle and the PCR products were purified with a GeneJET Gel Extraction Kit (Thermo Scientific, Waltham, MA, USA). Subsequently, the PCR product from each sample was sequenced on the Illumina MiSeq (300-bp paired-end reads) platform (Illumina Inc., San Diego, CA, USA) at the TinyGene Bio-Tech Co., Ltd. (Shanghai, China). The reads were distinguished from each sample and any sequences of low quality were deleted with the ultra-fast sequence analysis (USEARCH). The splicing sequence was qualified and filtered to yield the optimized sequences. Furthermore, operational taxonomic units (OTU) were obtained at a 97% similarity level using the UPARSE pipeline [24].

### 2.4. Data Analysis

Community diversity indexes, including Ace, Chao1, Shannon, and Simpson, were used to estimate bacterial abundance and diversity. The Chao1 index and Ace index indicated the richness of the bacterial community, and the Shannon index and Simpson index showed the diversity of the bacterial communities. An analysis of variance (ANOVA) test was conducted to assess the significance of tree characteristics and soil properties by IBM SPSS Statistics 26. The relationships of denitrifying bacteria with *A. spinulosa’*s growth and soil properties were determined using Spearman rank correlation analysis. Graphics were processed by Adobe Illustrator CC, version 2021, and Adobe Photoshop CC, version 2021.

## 3. Results

### 3.1. Effects of Invasion on Forest Structure and Soil Properties

In this study, we identified three sampling sites with different invasion degrees of *P. pubescens*, whose importance values were 0.0008 (low), 0.6551 (mod), and 0.9326 (high), respectively (Figure 1A–C). The biomass of *P. pubescens* roots under high invasion was significantly higher than that under moderate and low invasions (*p* < 0.01) and the results were consistent in the soil layers of 10 cm, 20 cm, and 30 cm (Figure 1D). At the same time, the biomass of *A. spinulosa* under moderate invasion was significantly higher than that at low and high invasions in the soil layers of 10 cm and 30 cm (Figure 1E, *p* < 0.01), suggesting that intermediate disturbance might contribute to the survival of *A. spinulosa*.

The rhizospheric soil physical and chemical properties of *A. spinulosa* were affected by different degrees of *P. pubescens* invasion. Notably, AN content in the high-invasive group was distinctively higher than that of the low-invasive group (*p* < 0.01) and slightly higher than that of the moderate group (*p* < 0.1) (Figure 2A). Nonetheless, no alterations were identified in pH, WC, or OM between the three comparisons (Figure 2B–D).

### 3.2. Effects of Invasion on Denitrifying Bacterial Community Composition

After sequencing, a total of 201,641 raw reads and 119,600 clean reads were collected from all nine soil samples. The number of sequences ranged from 36,918 to 43,944 per group, with a mean of 39,867 sequences. To avoid any bias in the distribution of taxa, the soil bacterial diversity of each sample was obtained according to rarefaction curves. Finally, we yielded a total of 5 phyla, 10 classes, 21 orders, 28 families, 32 genera, and 41 species of denitrifying bacterial taxa. Specifically, denitrifying bacterial sequences were primarily composed of the phyla Proteobacteria (27.53%), Bacteroidetes (5.26%), Firmicutes (1.61%), Cyanobacteria (1.21%), and Acidobacteria (0.81%). Additionally, Gammaproteobacteria dominated in all three groups, followed by Chitinophagia, Betaproteobacteria, and Alphaproteobacteria (Figure 3A). In addition, *Shewanella*, *Chitinophaga*, and *Achromobacter* were the primary genera found in three groups (Figure 3B). Notably, the top three genera found in the study showed the highest relative abundances in the high invasive group. The abundance in the moderate group was also higher than that of the low group (Table 2).

We then explored the shared and specific OTUs based on the Venn network (Figure 4). The results showed that a total of 22, 9, and 21 OTUs were identified as unique to low-, moderate-, and high-invasive groups, respectively. Additionally, a total of 136 OTUs existed in all samples, and most OTUs belonged to *Chitinophaga* (2.94%), *Shewanella* (2.21%), and *Stenotrophomonas* (2.21%). For those OTUs showed both in two groups, we found that 12, 22, and 24 OTUs were represented in low + moderate, low + high, and moderate + high groups, respectively. The clustering heatmap showed that moderate- and high-invasive groups were highly clustered, indicating that *P. pubescens* invasion could reshape the compositional structure of denitrifying bacteria.

### 3.3. Effects of Invasion on Denitrifying Bacterial Community Diversity

Diversity indexes, Ace, Chao1, Shannon, and Simpson were used to evaluate the diversity of soil denitrifying bacterial community in the rhizospheric soil of *A. spinulosa*. The mean coverage values in the three groups were larger than 0.999, indicating that the obtained OTUs could comprehensively evaluate the alterations of denitrifying bacterial diversity (Table 3). The Ace and Chao1 indexes in low- (190.18 and 191.16) and moderate- (205.49 and 204.50) invasive groups were lower than those of the high- (207.69 and 206.75) invasive group, and the two indexes increased with invasive degree. Similarly, the Shannon index was ranked as follows: high invasion (4.29) > moderate invasion (4.19) > low invasion (4.15). Contrarily, the ranking of the Simpson index was moderate invasion (0.030) > low invasion (0.029) > high invasion (0.024). We then analyzed beta diversity to further evaluate the similarities and differences of different invasive degrees on the denitrifying bacterial community. Weighted unifrac clustering trees based on taxonomy showed that moderate- and high-invasive groups were highly clustered at both the class and genus levels (Figure 5), which was consistent with the clustering heatmap described above.

### 3.4. Relationships of Denitrifying Bacteria with Soil Properties and the Growth of A. spinulosa

Spearman correlation analysis was performed on the soil properties and the denitrifying bacterial community abundance at the genus level (Table 4). The results showed that most denitrifying bacterial genera (e.g., *Shewanella*, *Chitinophaga*, and *Achromobacter*) were positively correlated with soil OM and AN but negatively correlated with soil pH and WC. Notably, soil OM was significantly altered by *Shewanella* (*R*^2^ = 1.000, *p* < 0.05), soil WC was altered by *Chitinophaga* (*R*^2^ = −1.000, *p* < 0.05), *Achromobacter* (*R*^2^ = −1.000, *p* < 0.05), and *Anaeromyxobacter* (*R*^2^ = −0.998, *p* < 0.05). When considering the growth of *A. spinulosa*, we found that *Chitinophaga* was distinctively correlated with height (*R*^2^ = −0.984, *p* < 0.05) and *Sorangium* was dramatically correlated with crown width (*R*^2^ = −0.999, *p* < 0.05).

## 4. Discussion

Plant invasion has distinctive impacts on plant characteristics and soil microbial communities [25,26]. Growing evidence has shown that microbial communities plays an essential role in inducing the successful expansion of invasive species. As a result, the physical and chemical properties of the colonized soil are altered, making it more suitable for the expansion of invasive species and further strengthening the situation of invasion. In addition, it has been demonstrated that soil N cycling can be an essential driver in forest succession [27,28]. Notably, altered total N content, ammonification rates (soil NH^4+^-N), and nitrification rates (soil NO^3-^-N) following the *P. pubescens* expansion into broad-leaved forests have been shown by many researchers in various areas [29]. Hence, the objectives of this study were to evaluate the invasive effects and mechanisms of bamboo on an endangered plant’s habitat and demonstrate the role of denitrifying bacteria on the successful expansion of *P. pubescens.*

### 4.1. Successful Moso Bamboo Invasion Altered Root Biomass and Soil N Content

In this study, we identified three moso bamboo invasive forests with low (0.0008), moderate (0.6551), and high (0.9326) IMP values. Successful invasions dramatically altered the underground root biomass of both *P. pubescens* and *A. spinulosa*. Specifically, the root biomass of *P. pubescens* was increased with the degree of invasion at the soil layers of 10 cm, 20 cm, and 30 cm. Bamboo has a capacity for carbon sequestration, owing to its fast growth and dense rooting system, dominating the soil layer of 0~20 cm [15,30]. Hence, *P. pubescens* can spread to other forestlands through the growth of roots and stems in the underground part, so that root biomass can be significantly increased [31]. Our findings are consistent with a previous study that reported the total root mass of an invasive species, *Imperata cylindrica* (cogongrass), increased at the 5–15 cm depth after invading longleaf pine (*Pinus palustris*) forests [32,33]. Interestingly, the root biomass of *A. spinulosa* was maximal at depths of 10, 20, and 30 cm under moderate disturbance. The intermediate disturbance hypothesis suggests that high species diversity can be maintained under moderate disturbance, and our findings proved that moderate disturbance of *P. pubescens* increased the root biomass of *A. spinulosa* and might have indirectly contributed to the growth and survival of the colonized plant [34,35]. This observation is also in agreement with a previous study that reported that the biomass harvesting of invaded *Populus tremuloides* and *Pinus Bansksiana* forests was greatest at intermediate disturbance severities [36].

The expansion of invasive plants can affect other plants’ rhizosphere soil’s physical and chemical properties [37]. Studies have demonstrated that the soil in Chishui Alsophila National Nature Reserve is acidic soil that permits the expansion of moso bamboo. Additionally, suitable OM content and WC help moso bamboo easily invade the original habitat of *A. spinulosa* [15]. These soil properties were not altered under different invasive groups, indicating they might not be the primary driver of the expansion of moso bamboo. Soil N accumulation primarily relies on N input through root mortality and N output by decomposition [38]. Hence, the increased soil AN under high disturbance was mainly due to an extensive underground bamboo root system and microorganisms, which entailed a highly dynamic N cycling process. These findings agree with other reports that the fluxes of N in the bamboo-dominant forest were 37.5% larger than that of a neighboring secondary, evergreen, broadleaved forest [29]. In addition, increased N uptake could prevent primitive tree growth, seed germination, and seedling establishment, thereby threatening plant biodiversity [2,29]. Taken together, the successful invasion of *P. pubescens* distinctively altered the N content in the rhizosphere soil of *A. spinulosa*, making it more suitable for the expansion of *P. pubescens*.

### 4.2. Moso Bamboo Invasion Increased Denitrifying Bacterial Diversity and Reshaped Its Composition

This study found that the invasion of *P. pubescens* exerted a distinctive effect on denitrifying bacterial community diversity of *A. spinulosa* rhizospheric soil. Based on the Ace and Chao1 indexes, the study identified that the community richness under moderate and high invasion was greater than under low invasion. Similarly, high invasion harbored the highest denitrifying bacterial community diversity, according to the Shannon and Simpson indexes. Similar findings illustrated that moso bamboo invasion of broadleaf forests increased soil fungal alpha diversity and community composition closely involved in soil carbon and N production [8]. Additionally, high *Rhynchelytrum repens* invasion and *Wedelia trilobata* invasion increased the associated bacterial diversity involved in nutrients uptake, thereby permitting a successful expansion into the native habitat [39,40]. Besides, the weighted unifrac tree supported our viewpoint that successful moso bamboo invasion showed distinctive denitrifying bacterial community structure compared with the low-invasive group. This finding is in agreement with a previous report that illustrated reed canary grass invasion increased the ratio of denitrifying to total bacteria [41]. Hence, the increased abundance and diversity of denitrifying bacteria entailed a well-developed N cycling process, which was consistent with the observation about AN content described above.

The bacteria with denitrification ability are distributed in more than 50 genera, most of which belong to Alphaproteobacteria, Betaproteobacteria, and Gammaproteobacteria [42]. *Shewanella* is a dominant denitrifying genus in Gammaproteobacteria and has been found in various habitats, such as oceans, lake sediments, and humid environments [43]. Hence, it is speculated that the denitrifying bacteria play essential roles in the Chishui Alsophila National Nature Reserve due to the mid-subtropical humid monsoon climate and well-developed river system [44]. *Shewanella* can maintain its own metabolism as a carbon source by fermentation products and can be utilized as electron acceptors, making it more viable in different ecosystem environments. In addition, *Chitinophaga* and *Achromobacter* were also the primary genera found in the study, and their abundances increased with invasive degree. Our observations are in line with a previous soybean study showing the abundance of *Chitinophaga* and *Achromobacter* changed with the addition of soybean residues and increased with exposure time [45]. Hence, the increased *Chitinophaga* and *Achromobacter* abundances were presumed to be due to the increasing *P. pubescens* litter residues, but further validation will strengthen and echo our hypothesis. In addition, the increase in denitrifying bacteria must necessarily be correlated with the increase in nitrogen-fixing bacteria, so future studies should investigate the alterations of nitrifying bacteria.

### 4.3. Relationships of Denitrifying Bacteria with Soil Properties and the Growth of A. spinulosa

The results of soil physical and chemical properties showed that the soil AN content was increased with *P. pubescens* invasion, which would inevitably promote soil N cycling and strengthen denitrification. In addition, denitrification, as an essential part of N cycling, can reduce nitrate in soil and alleviate the toxicity induced by nitrate accumulation [46,47]. According to the Spearman correlation analysis, the abundances of most denitrifying bacteria were positively correlated with soil OM and AN. The complex rhizomes and roots of *P. pubescens* could fill up the soil N and carbon pools, to some extent, and provide enough N and carbon sources for denitrifying bacteria, as described previously [21]. In addition, the decomposition of bamboo litter also increased the content of soluble carbon and nitrogen, which accelerated the nutrient cycling of soil, thereby increasing the abundance of denitrifying bacteria, such as *Shewanella*, and creating an invasive habitat for *P. pubescens* in turn. Interestingly, the abundances of *Chitinophaga*, *Achromobacter*, and *Anaeromyxobacter* were negatively correlated with WC. Growing evidence has shown that a high moisture content decreases rates of OM decomposition and that soil WC decreases microbial activity by altering the diffusion of soluble substrates, bacterial movement, and intracellular water potential [48,49]. Hence, soil WC might determine *C**hitinophaga*, *Achromobacter*, and *Anaeromyxobacter* structures by controlling nutrient availability and cell movement, which is in line with a previous study by Zhou et al. [50], who pointed out that soil WC connecting soil particles may influence soil bacterial diversity patterns.

Tree height and crown width are important indicators for photosynthesis and the respiration of trees, directly influencing the viability and productivity of plants [51]. Evidence has been found that bamboo invasion reduced mean tree height (11.7 m) compared with native broadleaved forest (13.1 m) [29]. Contrarily, certain invasive vegetation removal can increase basal diameter, tree height, and phylogenetic diversity of wetland plant communities [52]. Our Spearman correlation analysis showed that tree height and crown width of *A. spinulosa* were negatively correlated with two denitrifying bacteria, *Chitinophaga* and *Sorangium*, respectively. It is reasonable to assume that the high abundances of *Chitinophaga* and *Sorangium* promoted active N cycling suitable for bamboo expansion, whereas the bamboo expansion might further threaten the growth of the invaded plant *A. spinulosa*. Hence, *Chitinophaga* and *Sorangium* might be the putative indicators for evaluating the effects of bamboo invasion on the growth of *A. spinulosa.*

To better understand this putative mechanism of invasion by *P. pubescens*, it would be a priority to compare its rhizospheric soil denitrifying bacterial communities in its native and non-native habitats in future research, which can assist in estimating the enemy expansion and enhanced mutualism hypotheses. To further explore the abiotic and biotic responses of *A. spinulosa*, investigations of the diversity and richness of other microorganisms (e.g., nirS and nosZ denitrifying bacteria, nitrifying bacteria, azotobacteria, arbuscular mycorrhizal fungi, and ectomycorrhizal fungi) in natural/rhizosphere soils should be included in our future studies, as they are also crucial in participating in soil N cycles and plant growth and survival.

## 5. Conclusions

Our study found that low, moderate, and high *P. pubescens* invasions altered the underground root biomass of *P. pubescens* and *A. spinulosa*. Interestingly, we also observed that moderate invasion might play a positive role in the growth and survival of *A. spinulosa*. We further identified that bamboo invasion increased the soil N cycling of *A. spinulosa* rhizospheric soil with the help of denitrifying bacteria (e.g., *Shewanella*, *Chitinophaga*, and *Achromobacter*). The enhanced N cycling process in the rhizosphere soil of *A. spinulosa* made it more suitable for the expansion of *P. pubescens*. Futhermore, our observations illustrated that two denitrifying bacteria genera might be critical indicators for evaluating the effects of bamboo invasion on the growth of *A. spinulosa*. To sum up, this study was the first to analyze the effects of moso bamboo on the community structure of denitrifying bacteria in the colonized plant’s rhizospheric soil, providing essential contributions for the understanding of bamboo’s invasive mechanisms and the habitat protection of the *A.spinulosa* community.

## Figures and Tables

**Figure 1 microorganisms-10-00180-f001:**
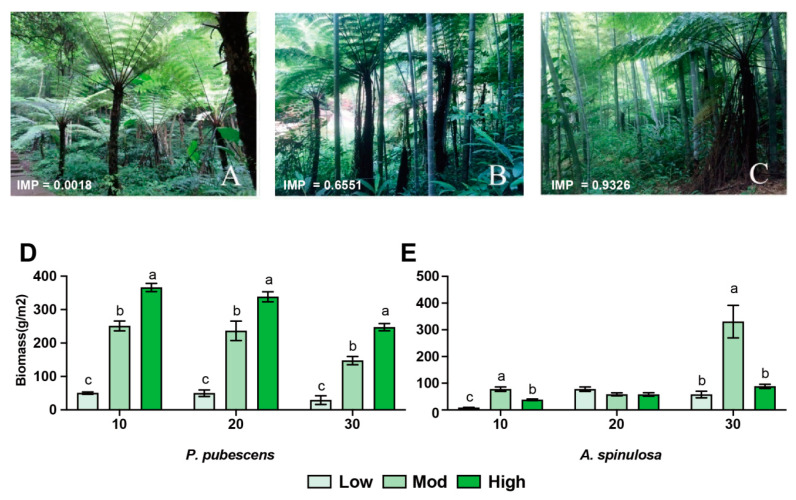
Effects of moso bamboo invasion on forest structure and tree characteristics. (**A**–**C**) Field pictures of *A. spinulosa* community disturbed by low (**A**), moderate (**B**), and high (**C**) moso bamboo invasions. IMP indicates the importance value of moso bamboo. (**D**,**E**) Root biomass of moso bamboo (**D**) and *A. spinulosa* (**E**) at different vertical depths, respectively. ^abc^ Different superscripts indicate the significant difference (*p* < 0.05).

**Figure 2 microorganisms-10-00180-f002:**
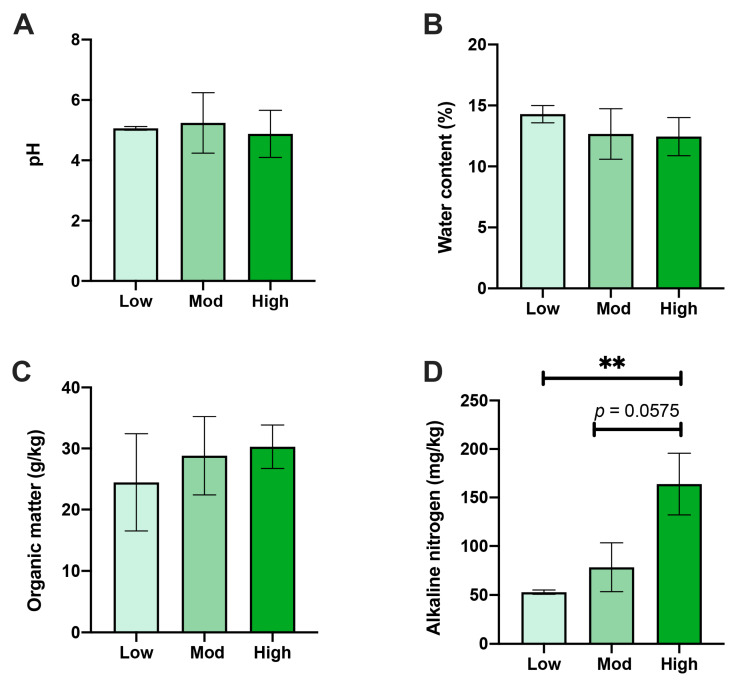
Effects of moso bamboo invasion on rhizospheric soil properties of *A. spinulosa.* Alterations of (**A**) soil pH, (**B**) water content, (**C**) organic matter, and (**D**) available nitrogen under low, moderate, and high moso bamboo invasions. ** Asterisk indicates the significant difference (*p* < 0.01).

**Figure 3 microorganisms-10-00180-f003:**
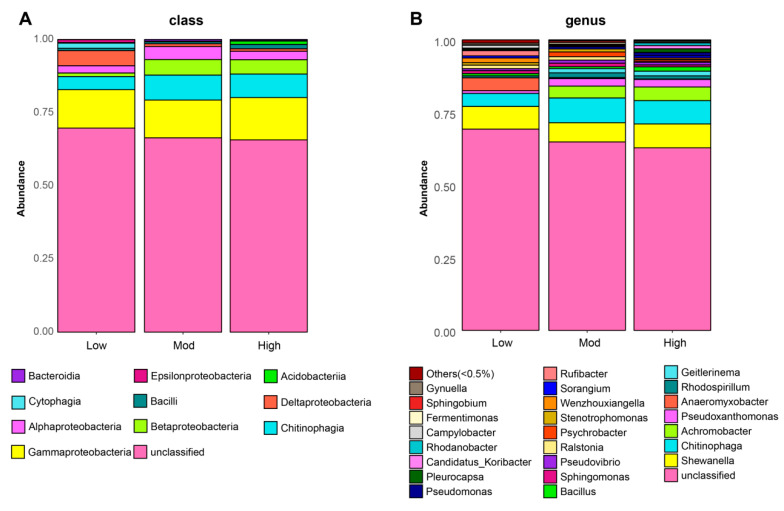
Effects of moso bamboo invasion on denitrifying bacterial community composition. Barplots showing the denitrifying bacterial community composition at class (**A**) and genus (**B**) levels.

**Figure 4 microorganisms-10-00180-f004:**
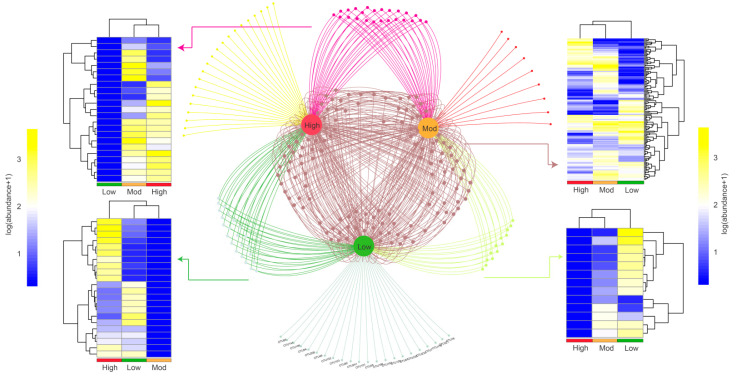
OTUs association network showing interactions between different treatments. The nodes represent OTUs shared or specialized by low, moderate, and high moso bamboo invasions. Heatmap showing the clustering patterns and abundances of OTUs identified in the three groups.

**Figure 5 microorganisms-10-00180-f005:**
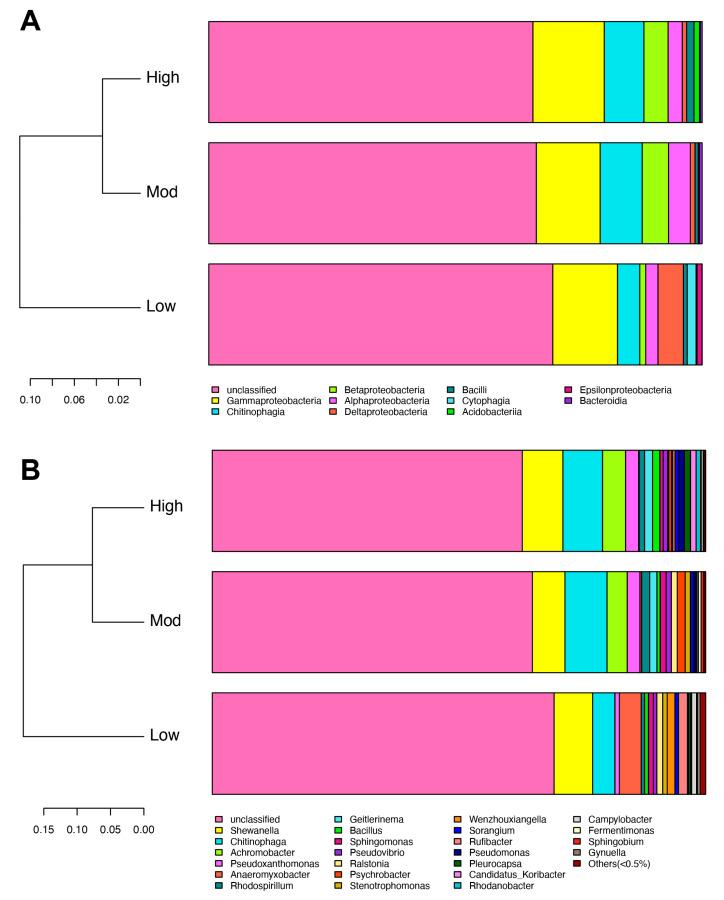
Effects of moso bamboo invasion on denitrifying bacterial community beta diversity. Barplots showing the denitrifying bacterial community composition at the class (**A**) and genus (**B**) levels. Weighted unifrac tree showing the distances of the identified denitrifying bacteria.

**Table 1 microorganisms-10-00180-t001:** The basic information about three sampling sites.

Intensities of Disturbance	Latitude and Longitude	Elevation/m	Importance Value	Dominant Woody Species
high	E 106°01′38.49″	564	0.9326	*Phyllostachys heterocycla*
	N 28°28′41.89″			
	E 106°38′22.30″	553		*Phyllostachys heterocycla*
	N 28°27′23.31″			
	E 106°01′37.36″	562		*Phyllostachys heterocycla*
	N 28°28′42.81″			
mod	E106°00′38.26″	547	0.6551	*Phyllostachys heterocycla, Alsophila spinulosa*
	N 28°27′38.00″			
	E 106°00′45.33″	525		*Phyllostachys heterocycla, Brassaiopsis glomerulata*
	N 28°19′40.14″			
	E 106°00′50.02″	527		*Phyllostachys heterocycla, Musa basjoo*
	N 28°25′36.44″			
low	E 105°58′00.83″	538	0.0008	*Casearia balansae, Alsophila spinulosa*
	N 28°23′39.04″			
	E 105°58′02.61″	535		*Lasianthus chinensis, Musa basjoo*
	N 28°23′38.55″			
	E 105°58′03.72″	523		*Musa basjoo, Alsophila spinulosa*
	N 28°23′37.46″			

**Table 2 microorganisms-10-00180-t002:** The relative abundance of dominant denitrifying bacteria in the rhizosphere of three sites.

Type of Bacteria	Low	Moderate	High	Coverage
*Shewanella*	0.066	0.078	0.082	0.076
*Chitinophaga*	0.045	0.08	0.085	0.07
*Achromobacter*	0	0.041	0.047	0.029
*Pseudoxanthomonas*	0.009	0.026	0.026	0.02
*Anaeromyxobacter*	0.044	0.003	0.001	0.016
*Stenotrophomonas*	0.009	0.002	0.01	0.007
*Sorangium*	0.007	0.007	0.006	0.007
*Campylobacter*	0.01	0.001	0.001	0.004
*Candidatus_Koribacter*	0.002	0.011	0.001	0.005
*Ralstonia*	0.012	0.002	0.012	0.009
*Fermentimonas*	0	0.004	0.006	0.003
*Bacillus*	0.008	0.015	0.007	0.01
*Sphingomonas*	0.01	0.007	0.012	0.01
*Rhodospirillum*	0.007	0.011	0.016	0.011
*Pseudovibrio*	0.007	0.009	0.011	0.009
*Sphingobium*	0.001	0.002	0.005	0.003
*Gynuella*	0.006	0	0.001	0.002
*Wenzhouxiangella*	0.016	0.005	0	0.007

**Table 3 microorganisms-10-00180-t003:** Diversity index of various groups of denitrifying bacteria.

Intensities of Disturbance	Ace	Chao1	Shannon	Simpson	Coverage
low	190.18	191.16	4.15	0.029	0.99971
moderate	205.49	204.5	4.19	0.03	0.99983
high	207.69	205.75	4.29	0.024	0.99959

**Table 4 microorganisms-10-00180-t004:** The relationships of denitrifying bacteria with plant and soil properties based on Spearman rank correlation analysis.

Type of Bacteria	pH	WC	OM	AN	Diameter	Crown Width	Height
*Shewanella*	−0.24	−0.99	1.000 *	0.835	0.343	−0.982	−0.999
*Chitinophaga*	−0.115	−1.000 *	0.991	0.758	0.221	0.555	−0.984 *
*Achromobacter*	−0.117	−1.000 *	0.992	0.759	−0.406	−0.706	0.984
*Pseudoxanthomonas*	0	−0.995	0.97	0.678	0.107	0.465	0.957
*Anaeromyxobacter*	0.041	−0.998 *	0.979	−0.708	−0.148	−0.492	−0.968
*Stenotrophomonas*	−0.918	0.3	−0.162	0.405	0.107	0.456	−0.114
*Sorangium*	0.866	0.588	0.696	−0.976	−0.915	−0.999 *	−0.73
*Campylobacter*	0	−0.995	−0.97	0.678	−0.107	−0.456	−0.957
*Candidatus_Koribacter*	0.908	−0.322	0.185	−0.383	−0.324	−0.617	0.138
*Ralstonia*	−0.866	0.407	−0.274	0.298	−0.871	−0.635	−0.228

Note: * Asterisk indicates the significant difference (*p* < 0.05).

## Data Availability

The data used to support the findings of this study are available from the corresponding author upon request.

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
