# Peer review of "Moso Bamboo Invasion Reshapes Community Structure of Denitrifying Bacteria in Rhizosphere of Alsophila spinulosa"

_microorganisms, 2022, doi:10.3390/microorganisms10010180_

Round 1

Reviewer 1 Report

I congratulate the work team for the paper.

is an interesting study and the data obtained are analyzed according to the statistical point of view.

please explain the graphs presented (some of which are not explained by example - Figure 1D and 1E; Figure 2 etc.

Author Response

Dear Professor (s),

Thank you very much for your letter dated on Jan 5, 2022. Based on your comment and request, we have made relevant modifications on the manuscript. Here, we attached a revised manuscript for your approval. A document answering every question was also summarized and enclosed.

A revised manuscript with the correction sections color marked was attached as the supplemental material for the purpose of easy checking and editing. Some grammatical or typographical errors have been corrected. All the lines and pages indicated above are in the revised manuscript.

Should you have any questions, please contact us without hesitate.

Thank you and all the reviewers for the kind advice.

Sincerely yours,

Hong-ping Deng

Reviewer comments:

Reviewer #1: I congratulate the work team for the paper. It is an interesting study and the data obtained are analyzed according to the statistical point of view. please explain the graphs presented (some of which are not explained by example - Figure 1D and 1E; Figure 2 etc.

Answer: We appreciate your constructive advice! Following your suggestions, we have elaborated further on the graphs presented in the manuscript.

Changes: Line 162-163 “in the soil layers of 10 cm and 30 cm” was added.

Line 273-275 “in the soil layers of 10 cm... cm (Qu et al., 2020” was added.

Line 295-297 “These soil properties...expansion of moso bamboo” was added.

Line 321-326 “Besides, the weighted unifrac tree...with low invasive prairie (Gordon ” was added.

Acknowledgements

We acknowledge the reviewer’s comments and suggestions very much, which are valuable in improving the quality of our manuscript. Should you have any questions, please contact us without hesitation.

Thank you for the kind advice!

Sincerely yours,

Hong-ping Deng

Reviewer 2 Report

In general, the article is more or less well written. But I would recommend to describe experimental design in more detail, especially the number of samples and their type.

Line 15 the sentence is unclear

Line 81 Aim 1 - should be rewritten " reveal how properties of soil colonized with A... "

Aim 2 - what does mean "critical" in this context?

Line 98 authors should explain the term "importance value"

Line 102 how many samples from different depths were taken? Were they mixed to get the average sample?

Line 109 how many rhizospheric samples were taken? Were there samples from different species?

Line 159 figure 1 was with the information about three depths. What samples were used for the analysis of physical/chemical properties? Are they also from 3 different depths? These should be clarified

Line 165 The information on figure and in the figure caption is not matching

Line 169 how many samples were sequenced?

Line 250 (high) should be changed to the number

Line 275 check spelling "bamboo"

Line 297 alphaproteobacteria is written twice, should be corrected

Line 317, 339 correct to Spearman

Line 329 correct to Chitinophaga

Line 357 it was not hypothesized, but it was observed

Author Response

Dear Professor (s),

Thank you very much for your letter dated on Jan 5, 2022. Based on your comment and request, we have made relevant modifications on the manuscript. Here, we attached a revised manuscript for your approval. A document answering every question from the referees was also summarized and enclosed.

A revised manuscript with the correction sections color marked was attached as the supplemental material for the purpose of easy checking and editing. Some grammatical or typographical errors have been corrected. All the lines and pages indicated above are in the revised manuscript.

Should you have any questions, please contact us without hesitate.

Thank you and all the reviewers for the kind advice.

Sincerely yours,

Hong-ping Deng

Reviewer comments:

Reviewer #1: I congratulate the work team for the paper. It is an interesting study and the data obtained are analyzed according to the statistical point of view. please explain the graphs presented (some of which are not explained by example - Figure 1D and 1E; Figure 2 etc.

Answer: We appreciate your constructive advice! Following your suggestions, we have elaborated further on the graphs presented in the manuscript.

Changes: Line 162-163 “in the soil layers of 10 cm and 30 cm” was added.

Line 273-275 “in the soil layers of 10 cm... cm (Qu et al., 2020” was added.

Line 295-297 “These soil properties...expansion of moso bamboo” was added.

Line 321-326 “Besides, the weighted unifrac tree...with low invasive prairie (Gordon ” was added.

Reviewer #2:

Q1: Line 15 the sentence is unclear

A1: Thank you for your helpful comments! We have rephrased the sentence in the abstract to declare a more precise context.

Changes: Line 14-17 “The uncontrolled invasion... the habitat of A. spinulosa.” was rephrased.

Q2: Line 81 Aim 1 - should be rewritten " reveal how properties of soil colonized with A... "

A2: Thank you! Corrected.

Q3: Aim 2 - what does mean "critical" in this context?

A3: Thank you! We have corrected this word into “dominant” accordingly.

Q4: Line 98 authors should explain the term "importance value"

A4: We appreciate your kind suggestion and have added a sentence to explain the term “importance value”.

Changes: Line 103-104 “The importance value (IMP) was...frequency of all species.” was added.

Q5: Line 102 how many samples from different depths were taken? Were they mixed to get the average sample?

Q6: Line 109 how many rhizospheric samples were taken? Were there samples from different species?

Q7: Line 159 figure 1 was with the information about three depths. What samples were used for the analysis of physical/chemical properties? Are they also from 3 different depths? These should be clarified

Q8: Line 169 how many samples were sequenced?

A5-A8: Thank you for your helpful comments! Three replicates for each depth were taken in the study to determine the biomass of root samples. To avoid heterogeneity, each sample was independent and not mixed to get the average sample.

The number of rhizospheric samples used for sequencing and properties determination was also clarified in the methods section. These samples were also independent.

The information about the depth of soil samples was clarified in the methods and discussion section. Again, we appreciate your careful work and apologize for our honest mistakes.

Changes: Line 111 “(three replicates)” was added.

Line 117-118 “(three replicates per group...0~20 cm) ” was added.

Line 120-121 “ Finally, a total of nine soil...three invasive groups” was added.

Line 184 “from all nine soil samples.” was added.

Line 273-275 “Bamboo has a capacity for ...the soil layer of 0-20 cm (Qu..” was added.

Q9: Line 165 The information on figure and in the figure caption is not matching

A9: Thank you for your helpful comments! We assume that the MDPI submission system may trigger this, and we will avoid this mistake in the revised manuscript.

Q10: Line 250 (high) should be changed to the number

A10: Thank you! Corrected.

Q11: Line 275 check spelling "bamboo"

A11: Thank you! Corrected.

Q12: Line 297 alphaproteobacteria is written twice, should be corrected

A12: Thank you! Corrected.

Q13: Line 317, 339 correct to Spearman

A13: Thank you! Corrected.

Q14: Line 329 correct to Chitinophaga

A14: Thank you! Corrected.

Q15: Line 357 it was not hypothesized, but it was observed

A15: Thank you! Corrected.

Acknowledgements

We acknowledge the reviewer’s comments and suggestions very much, which are valuable in improving the quality of our manuscript. Should you have any questions, please contact us without hesitation.

Thank you for the kind advice!

Sincerely yours,

Hong-ping Deng